# A New Post-Processing Proposal for Improving Biometric Gait Recognition Using Wearable Devices

**DOI:** 10.3390/s23031054

**Published:** 2023-01-17

**Authors:** Irene Salvador-Ortega, Carlos Vivaracho-Pascual, Arancha Simon-Hurtado

**Affiliations:** Departamento de Informática, Escuela de Ingeniería Informática de Valladolid, Universidad de Valladolid, Paseo de Belén 15, 47011 Valladolid, Spain

**Keywords:** gait recognition, smartwatch, accelerometer sensor, window fusion technique, cross-session tests

## Abstract

In this work, a novel Window Score Fusion post-processing technique for biometric gait recognition is proposed and successfully tested. We show that the use of this technique allows recognition rates to be greatly improved, independently of the configuration for the previous stages of the system. For this, a strict biometric evaluation protocol has been followed, using a biometric database composed of data acquired from 38 subjects by means of a commercial smartwatch in two different sessions. A cross-session test (where training and testing data were acquired in different days) was performed. Following the state of the art, the proposal was tested with different configurations in the acquisition, pre-processing, feature extraction and classification stages, achieving improvements in all of the scenarios; improvements of 100% (0% error) were even reached in some cases. This shows the advantages of including the proposed technique, whatever the system.

## 1. Introduction

Biometric recognition is an important field for both economics (with a market in continuous growth) and research, where the search for and study of new alternatives is a question of continuous interest. From these, those related to wearables have aroused great interest in the biometric community in recent years due to the popularization of these devices. It can therefore be considered an emergent biometric technology with a promising future [1].

In this work, user authentication, or verification is approached by means of the behavioral biometric characteristic gait using a smartwatch. Biometric recognition encompasses two different tasks: verification and identification. In verification, the goal is to authenticate the user (Am I the person I claim to be?). In identification, the goal is to find the owner of the characteristic (To which person does it belong?).

Gait, as a behavioral biometric characteristic [2] like speech, signature, or keystroke, is based on *measurements and data derived from an action of the user*, in this case, the user’s personal movement while walking. That is, the goal in biometric gait recognition is to characterize individuals by their way of walking.

The use of gait shows interesting advantages with regard to other more mature biometric characteristics (generally, physiological ones, e.g., fingerprint, face or iris): the capture is unobtrusive (the user does not need to carry out a special action, only a common one as is walking) and it is also difficult to steal or falsify. In addition, with the appearance of wearables, its capture can be performed continuously (while the user wears the device), and it is easy to obtain by simply accessing the corresponding sensors.

The use of gait for user recognition has been posed since the beginning of the research in biometric recognition [3]. The main problem, until the appearance of smartphones, was that dedicated devices were used to capture the user’s way of walking, such as cameras or floor sensors [3]. In addition, its performance has usually been lower than other behavioral characteristics, for example, speech or signature.

A first step to make gait acquisition easier with the use of smartphones, more specifically, through the use of their inertial sensors (accelerometer and gyroscope). The use of these devices prevents the need to use external or specialized ones to capture user gait, such as the above-mentioned cameras or floor sensors. However, the signal acquired with these devices has the problem of being body-place-dependent [3,4]; i.e., the signal is different when the mobile is located, for example, on the belt, in the trouser pocket, or in the hand.

The appearance of Wearables solved this problem, since they are worn on a specific part of the body (head, neck, wrist, etc.), which is always the same.

With the popularization of these devices, the biometric community has shown considerable interest in them, due to their ability to capture both physiological and behavioral personal data, their use not being restricted to gait recognition. Some examples of these different approaches are heart-based authentication (using PhotoPlethysmoGraphy sensors [5] or Electrocardiogram [6,7]), speech [8,9], keystroke [10], galvanic skin response [11], and non-volitional brain response [12].

Nevertheless, gait has a major advantage with respect to others: it is easy to collect, as all current wearables have inertial sensors (accelerometer and gyroscope), these being the signals usually used for gait recognition [13]. Therefore, it is very suitable for general practical applications, which has increased the interest of the gait research community in these devices in recent years [3,4]. Here, our interest focuses on wrist wearables, since they are very popular, increasing their real usage possibilities.

From the two inertial sensors, accelerometer and gyroscope, the first is the most commonly used and the first referenced [14,15,16]. However, the gyroscope was rapidly incorporated [17]. From comparative works performed [18,19,20,21], the accelerometer has shown the best performance in general, so it is the one used here.

The systems proposed in the literature are based mainly on the standard biometric system [22,23] (Section 2) schematized in Figure 1. To improve recognition, different alternatives in the pre-processing, feature extraction, or classification stages have been proposed, as seen in Section 2. Here, we address the problem with a different approach, proposing the introduction of a post-processing stage between classification and decision making (Figure 2) where the Window Score Fusion Post-Processing (WSFPP) proposal is applied.

This proposal is based on several previous ideas:
*Fusion of features* that can be implemented at distinct levels and used in biometrics to improve accuracy [24]. Here, we use score-level fusion. Two main approaches can be found at this level: fusing scores (outputs) of different classifiers over the same input (e.g., [25]) and fusing the scores of different characteristics, *multibiometric system* (e.g., [26,27]). We propose a different and simpler approach here, since the scores of a single classifier and characteristic (gait) are combined (Section 4).*Exploiting context* to enhance system accuracy [28]. This is habitual in problems where the context is relevant, e.g., those related with video (e.g., [29]), prosody recognition (e.g., [30]), or speech. Since, in gait, a step is related to the ones that go before and after, our proposal aims to exploit this dependence (Section 4).

When analyzing the literature, only a few works were found that used post-processing, but they do not exploit the context and are not performed at the decision level (Section 2).

To test our proposal, a strict biometric testing protocol has been followed, taking reproducibility principles into account in the experimental protocol description. Testing is performed under a cross-session scenario: the data acquired for training and testing are from different days, which this is important in order to include the intra-user variability of the biometric traits [31] and to consider the “template ageing” problem [22], essential in real systems.

Besides the main contribution shown, two new and original proposals for signal cleaning and for gait cycle extraction (gait cycle is defined in Section 3) are set out. Furthermore, from the different comparisons carried out (Section 5.1), the time domain vs. frequency domain in the feature extraction (the two main approaches) has been included, which has not been included in previous work to the best of our knowledge.

The rest of the paper is organized as follows. In Section 2, the state of the art in gait recognition by means of wearables is analyzed, focusing on the configurations of the systems proposed. After describing the reference biometric system (a state of the art system) in Section 3, our novel WSFPP proposal is set out in Section 4. The experimental methodology can be seen in Section 5. The results are shown in Section 6, followed by their discussion in Section 7. To finish, the conclusions are given in Section 8.

The above shows the main narrative of the work, which is complemented with the content included in two Appendices: an original proposal for gait cycle extraction in Appendix A and another for signal cleaning in Appendix B.

## 2. Related Works

Different approaches to biometric recognition using wearables can be found in the literature [5,6,7,8,9,10,11,12]. However, those based on gait can be considered one of the most important and popular due to its ease of acquisition: all current wearables have inertial sensors, which are the signals usually used for gait recognition [13].

From the many related works that can be found, only a very small number include post-processing stages. Of these, to the best of our knowledge, none have proposed a technique similar to our proposal (Section 4). We have found two main related approaches, both based on *Fusion of features* [24]:
At the score level, using scores of different classifiers [32,33].At decision-level, with the voting scheme being the most frequently used [18,19,32].

Another important consideration is that most of the works about wearables and gait recognition use non-commercial devices. This can be seen in the public databases available: OU-ISIR [34], ZJU-GaitAcc [35], HuGaDB [36], or that acquired in [37], where devices built by the authors or simulated by means of Wii Remote are used. An exception can be found in the most recent WISDM dataset [18], but with the limitation of having only a single acquisition session.

From our point of view, the use of non-commercial wearables makes it difficult to extrapolate the results to real applications, so we are interested in works that use smartwatches or smartbands to test our proposal, since these are the ones most closely related to our approach. From a review of the literature, the following works using commercial wearables were found: [18,19,20,21,38,39,40,41,42,43,44,45,46].

In [20,39,43,46], the task addressed is identification, not verification. The approach of both problems is different, even the error measure is different. Here, we are interested in verification, and only the works that address this problem are used as reference.

Taking into account the aforementioned, Table 1 gives a brief description of the state of the art in the problem approached. In this table, besides the works that use commercial wearables, some recent and relevant ones that do not use these have been added to show a more complete view of the current state of the topic.

## 3. State-of-the-Art (Reference) System

In this section, we present a state-of-the-art biometric system (Figure 1) in gait recognition by means of wearables, which will be used as reference to test our proposal. This system is based on the most relevant related literature (review shown in the previous section).

Before continuing, it is important to define two concepts to be used from now on:
**Biometric sample**, or **sample** for short: the analog or digital representation (Figure 3) of biometric characteristics [51] (Figure 1), gait in our case. This is specified in more detail in Section 5.2.**Gait cycle**, or **cycle** for short: time period from when one foot contacts the ground until the same foot again contacts the ground (Figure 4). Therefore, the sample is a periodic signal where each period corresponds to a cycle (Figure 3).

For greater readability, some parts are described in the final appendixes.

### 3.1. Acquisition

While the user is walking, the raw signal from each coordinate (X, Y and Z) of the accelerometer sensor of the wearable is captured. This signal is divided into cycles, approaching this operation in a different way here from what appears in the literature. Our original proposal is shown in Appendix A.

### 3.2. Preprocessing

In this stage, the biometric sample is adapted and enhanced to improve its performance in the following stages. This stage encompasses the following tasks:
**Sample cleaning,** which consists of eliminating the noisy parts of the signal and detecting and correcting acquisition errors. Not much work has been carried out into acquisition problems with real devices, so this has led us to propose our own alternative, which can be seen in Appendix B.**Period Normalization.** With real devices, it is not possible to ensure a fixed sampling rate. This can be seen in Figure 5, where the distribution of the time between two consecutive datum of our database is shown. To fix this, the sample must be resampled [13,21]. To perform this operation, the following must be set: (i) the interpolation method and (ii) the sampling rate. For the first, and following the literature [13], linear interpolation has been used. For the second, after analyzing the frequency components of the data, we saw that components bigger than 6 Hz were negligible; so, following the Nyquist–Shannon sampling theorem, a sampling rate of 12 Hz (a period of 83.3 ms) was fixed. This value is in accordance with that shown in [52], where it is demonstrated that the arm moves at a maximum of 8.6 Hz, making the movement as fast as possible. As our data are collected from walking, the sampling rate selected seems reasonable.**Amplitude Normalization.** The goal is to change the value of the data to a common scale [13]. The need to perform this operation is machine learning algorithm dependent, so the default option for each classifier in the software used for the experiments (RStudio) is used.**Filtering** to soften the signal. Here, one of the most commonly employed algorithms [13,45,53], the Weighted Moving Average (WMA), has been used. This algorithm is defined as: $wma=\frac{d_{t-1}+d_{t}+d_{t+1}}{3}$, with $d_{t},d_{t-1},d_{t+1}$ being the data at the instant *t*, the instant before and the instant after, respectively.

### 3.3. Feature Extraction

The output of this stage is a mathematical representation of the biometric sample, suitable to be processed by a learning algorithm. For this, as can be seen in Table 1, most of the works propose to extract features in the time domain. However, since the sensor signal is a time series, feature extraction in the frequency domain has also been suggested [50,54]. Here, we test and compare the results with both.

The feature extraction from a sample is accomplished as follows:The cycles of the sample are grouped into segments called windows, with 20% of overlap [45] (Figure 3). Therefore, a window, $w_{j}$, is composed of *m* consecutive gait cycles. A cycle is a piece of signal that is too short to be representative of the user’s gait, so these are grouped into windows, which, from now on, will be the basic unit of information used to model and recognize the user.From each window, $w_{j}$, a feature vector, $F_{j}$, is extracted as shown in the next two sections.

#### 3.3.1. Time Domain Feature Extraction

Following the literature [2,18,45,47,55], the following features are extracted: mean, median, maximum, minimum, standard deviation, maximum range (maximum–minimum), kurtosis, 25th percentile, 75th percentile, asymmetry coefficient, energy, and maximum value of the autocorrelation.

#### 3.3.2. Frequency Domain Feature Extraction

First, the Fast Fourier Transform (FFT) is applied to each window to convert the signal to a representation in the frequency domain. After eliminating the zero component, the following features are extracted: the same statistics as in the time domain plus the maximum amplitude, the second maximum amplitude, the first dominant frequency, the second dominant frequency, and the under curve area.

### 3.4. Classification

Biometric verification is a binary classification problem, where the goal is to classify input data as belonging or not belonging to a certain target class, in our case, the claimant (subject to be authenticated) identity. For this, following what performed in biometrics, a different classifier, $\lambda_{C}$, is trained for each subject or claimant *C*. Its output (score), $s(F/\lambda_{C})$, with *F* as the output of the previous stage, will be a measure of the degree of belonging of the biometric sample to the user.

As can be seen in Table 1, a wide variety of machine learning algorithms of diverse types have been used. From these, the most used (used in three or more works) are the Support Vector Machine (SVM), Multilayer Perceptron Artificial Neural Network (MLP), K-Nearest Neighbor (K-NN), and Random Forest (ensemble method based on Decision Trees).

Except for K-NN, the rest of the classifiers must be trained with samples of the claimant and examples of the “non-claimant” class, i.e., “other” subjects, which in biometrics is called the impostor class. The data used for this classifier training are described in Section 5.3.

## 4. Window Score Fusion Post-Processing Proposal

Our system proposal is shown in Figure 2, where the new post-processing stage for the window score fusion is added to the *reference system*.

The classifiers proposed, both here and in the bibliography, have the problem that the temporal relation between windows is lost, i.e., that each window is treated separately. In a temporal series, as the gait is, it is normally important to capture the relations between consecutive events (gait cycles in our case). This is the goal with the technique proposed and described in this section, where this relation is “captured” at score level.

The scores (classifier outputs, Figure 2), $s_{j}=s\left(F_{j}/\lambda_{C}\right)$, of several *n* consecutive windows are fused to obtain the final output of the system (Figure 6) as shown in Equation (Equation 1).
(1)$$s_{l}^{*}=f\left(s_{k},s_{k+1},\ldots ,s_{k+n-1}\right)$$
where $s_{j}=s\left(F_{j}/\lambda_{C}\right)$ is the output of the classifier $\lambda_{C}$ for the feature vector $F_{j}$ extracted from the window $w_{j}$ and f() is the fusion function. Now, $s_{l}^{*}$ will be the measure of the degree of belonging of the biometric sample to the user.

Focusing on the fusion function, f(), we can find many options in the bibliography [32,56]. Here, fusion techniques based on weighting each element to be fused (e.g., weighted sum [56]) are not suitable, since we have no prior knowledge of their values or advantages because the scores to be fused come from the same classifier. Moreover, for practical considerations, simplicity is advisable, so complex fusion techniques, e.g., [57], were discarded. Thus, based on the above and our own experience [56,58], we selected the following simple fusion techniques: mean, max, min, and median.

## 5. Experimental Methodology

### 5.1. Experiment Design

The goal with the experiments is to prove that the WSFPP proposal improves the recognition rates independently of the system configuration, i.e., to demonstrate that the WSFPP performance does not depend on the previous stages of the system, so it is suitable for any system.

For this, we start from the review of the state of the art shown in Section 2 and Section 3. From this review, we select the following system/experiments configuration to test our proposal:
With regard to the **sensor coordinate** in the **acquisition stage**, the performance of using each sensor coordinates (X, Y and Z) separately, or fusing all by means of the *module* [27,59] ($Module=\sqrt{X^{2}+Y^{2}+Z^{2}}$), is calculated and compared.With regard to the **window size** in the **pre-processing stage**, from the values in Table 1, the following number of cycles in the window were selected to be tested: {2,4,8,12}. This set includes both small and high values, being representative of those used in the state of the art.With regard to the **features** extracted in the **feature extraction stage**, as shown in Section 3.3, typical features are extracted in both the time and frequency domains. In addition, this allows the performance of both approaches to be compared.With regard to the **classifier** in the **classification stage**, two criteria were fixed:Variety in the tested algorithms.Most used in the reference works.The most used were shown in Section 3.4: SVM, MLP-ANN, K-NN, and Random Forest. These also fulfill the first criterion, since their theoretical bases are completely different. A deep study of each classifier is beyond the scope of this work, so a brief description of each one is included, focusing on the main differences between them:**–** **K-NN.** Unlike the rest of the classifiers, this does not need to be trained to build a model. The training or enrollment sample(s) (see Section 5.3) is (are) used directly to create the user template, $\lambda_{C}$. More specifically, the user template is made up of the feature vectors extracted from the enrollment sample(s): $\lambda_{C}=\left\{F_{i}^{e}\right\}\;0\leq i\leq N$, where *N* is the number of windows of the enrollment sample(s). The classifier output is based on distance; to be precise, given a test trial feature vector $F_{j}^{t}$ (see Section 5.3), its score is calculated as shown in Equation (Equation 2).
(2)$$s\left(F_{j}/\lambda_{C}\right)=Mean\left(kmin_{i}\left(EuclideanDistance\left(F_{j}^{t},F_{i}^{e}\right)\right)\right)$$Since the K-NN output is based on distance, the $s(F/\lambda_{C})$ interpretation is as follows: the lower its value, the higher the degree of belonging of the biometric input to the user.**–** **SVM** [60]. This classifier is based on separating two classes by means of an optimal hyperplane ${\vec{w}}^{T}\vec{x}=0$. The parameters of the hyperplane are fixed using the so called “support vectors” (Figure 7a). To avoid overfitting, a soft margin solution is used in the training phase (calculation of hyperplane parameters), allowing “mistakes” in the training samples (Figure 7a); this is controlled by the regularization parameter *C*: a small *C* allows a large margin of mistakes, while a large *C* makes constraints hard to ignore. With the hyperplane set, the classification is performed as shown in Equation (Equation 3).
(3)$$\begin{array}{c} {\vec{w}}^{T}\vec{x}\geq 0\Rightarrow Class\;A\\ {\vec{w}}^{T}\vec{x}<0\Rightarrow Class\;B \end{array}$$The problem is that real-world data are rarely linearly separable. The solution is to increase the dimensionality of the feature space, aiming to map the input space into a linear separable one, where the linear classifier will be applicable. This is performed by means of the “kernel trick”; i.e., a kernel function (e.g., lineal: k(x,y)=x·y, radial: $k\left(x,y\right)=e^{\gamma {∥x-y∥}^{2}}$) is used, allowing the mapping to be performed without increasing the complexity of the training algorithm.As can be seen in Equation (Equation 3), the sign of the output is used to classify the input. However, here we need a score, i.e., a level of belonging to each class. The Platt scaling [61] is used to accomplish this. Therefore, the score here is a probability, $s\left(F_{j}/\lambda_{C}\right)=P\left(F_{j}/\lambda_{C}\right)$. Therefore, with a different interpretation regarding K-NN, the higher its value, the higher the degree of belonging of the biometric input to the user.**–** **MLP** [62]. This is a net composed of a set of neurons or units organized in layers (Figure 7b). The architecture of the net is defined by the number of layers and neurons in each layer. Each neuron in a layer is connected (its outputs are the inputs) with all the neurons of the following layer, except the last one, whose neurons will be the output(s) of the net. The first layer is the input of the net, which will be the feature vector. The operation performed for each neuron is that shown in Equation (Equation 4), where $y_{h}^{p}$ is the output of the neuron *h* for the input *p*, $w_{jh}$ is the weight (real number) that connects the neuron *h* with the neuron *j* of the previous layer, $y_{j}^{p}$ is the output of this neuron *j* for the input *p*, and $\theta_{h}$ is the *bias* or *offset* of the neuron *h*. F is a function that must be derivable; typical functions are the sigmoid or the hyperbolic tangent.
(4)$$y_{h}^{p}=F\left(\sum_{j}w_{jh}y_{j}^{p}+\theta_{h}\right)$$During the learning or training stage, the weights of all the neurons are set using the backpropagation algorithm, so that the value of an error function, *E*, will be minimized. The most common error function is the squared error, $E=\frac{1}{2}{\left(d_{o}^{p}-y_{o}^{p}\right)}^{2}$, where $d_{o}$ is the desired output for the output neuron of the net *o* for the input *p* and $y_{o}$ is the output of the neuron.In our problem, the net has a single neuron in the output, being trained to obtain 0 (the desired output) for training examples of the impostor class and 1 for training examples of the subject (authentic class), using the sigmoid as the activation function, F. Therefore, for the MLP, $s\left(F_{j}/\lambda_{C}\right)=y_{o}^{F_{j}}=F\left(\sum_{j}w_{jo}y_{j}^{F_{j}}+\theta_{o}\right)$, with *j* being each neuron of the last hidden layer (l−2 in Figure 7b). Although the output is not really a probability, due to the values of the desired outputs, it can be considered as such, so its interpretation is the same as that seen with SVM.**–** **Random Forest** [63]. This is an ensemble of relatively uncorrelated *decision tree* classifiers. A decision tree is a supervised classifier that has a flowchart-like tree structure (Figure 7c); each *decision node* represents a decision rule, finishing in the *leaf nodes* with the final decisions. This tree is constructed following the algorithm below:Using Attribute Selection Measures (ASM), select the best feature (attribute) to split the dataset.Create the decision node with the corresponding decision rule. If the node is the first, it is called the *root node*.Using the decision rule, divide the corpus into subsets.Repeat the previous steps recursively for each subset until the nodes cannot be further classified due to all of the subset belonging to the same feature value, due to there being no more features, or due to there being no more data.Based on this classifier, Random Forest works as follows:*Training stage:Split the training set randomly into subsets with replacement.Train a decision tree with each subset.*Prediction or test stage:Each tree predicts a class.Probabilities are calculated from these classes using the predictions.Therefore, for this classifier, $s\left(F_{j}/\lambda_{C}\right)=P\left(F_{j}/\lambda_{C}\right)$.These classifiers were those selected for the tests. In addition to the above, as will be seen in the results, the performance of the selected classifiers is very different, which confirms the variety of the selections.For their configuration, we tried to use the default options of the software used (RStudio) as much as we could; the reason is to avoid possible bias in the results when optimizing the classifier, since our goal here is not to obtain the best results, but to test our proposal in the most objective way. Under this criterion, only the following particular configurations were posed:–SVM: radial kernel. From previous experiments, this kernel showed the best performance. R library used: e1071.–MLP-ANN: *JE_Weights* initialization function was selected. Others were tested, but the system showed inconsistencies. R library used: RSNNS.–K-NN: k=1 was selected. As with the SVM, from previous experiments, this value showed the best performance, and it is the simplest configuration. R library used: FNN.–Random Forest: no particular configuration was used in this case. R library used: randomForest.

With regard to the WSFPP technique, the following values of *n* (number of consecutive windows fused) were tested: n={2,4,8}. From the different fusion functions proposed, we decided, for clarity in the exposition, to select one: as can be seen in the next section, the results are clear, and adding more comparisons would not, in this case, provide new information and would only complicate the reading of the results. From the preliminary experiment, we saw that min and max performed worse than median and mean. The performance of these last two was similar, but median was slightly better, so this was chosen as the fusion function to test our proposal.

To achieve the goal posed, the following procedure was used:The performance of the reference system (Section 3) was calculated in all of the proposed system configurations.The performance, when our proposal was used (Section 4), was also calculated in all of the proposed system configurations, and for the different values of *n*.Both results, under the same system configurations, were compared.For objective results, the experimental conditions were the same in all of the experiments performed.

As can be seen (and as can be seen in the results), we compared the performance of using and not using (i.e., the reference system) our proposal in 128 different scenarios (system configurations), and for each one, with three different configurations (values of *n*) of our proposal.

### 5.2. Corpus Data Acquisition

The biometric data were captured by means of a Motorola Moto 360 watch (the same smartwatch as in [44]). This device has several sensors, and the data from all of them were collected to be used in future research. In this work, we focus on data from the 3D-accelerometer.

The data were acquired in a normal walking scenario [37,45], the most interesting one in our opinion, since it is a very common daily activity.

To study dependence over time, two different sessions were held, with a minimum separation of two weeks between them; in most cases, the separation was greater than one month, even two to three months for several subjects. As can be seen, the database includes a wide range of time intervals between sessions.

In each session, the subjects walked twice with the device on their wrist, on average 4 min for each walk, with a rest of about three to five minutes between them; therefore, in each session, two *biometric samples* were captured. The walks were outdoors, in different places. The place, walking speed, and type of footwear or clothing between sessions were not controlled; the only variable that was controlled was that the surface was flat (road or sidewalk).

In the end, data from 38 volunteer subjects, 25 men and 13 women, with a wide age range from 16 to 57 years, were captured. As indicated above, each sample consists of the 3D-accelerometer data collected.

### 5.3. Experimental Sets

Two main scenarios can be considered with the corpus:
**Short period authentication**: the enrollment and testing samples belong to the same session.**Long period authentication**: the enrolment and testing samples belong to different sessions. Testing under this condition is critical as user behavior is different from day to day.

In this work, we only examine the second scenario, as, although less favorable, it is the most realistic. This conditions the division of the database into training and testing, as shown in the following.

We train a different classifier, $\lambda_{C}$, per each subject *C* in the database using:**Enrollment data** (genuine class training set). “Enrollment” is, in biometrics, the step where a subject (claimant) *C* supplies the biometric data to build their biometric model or template, $\lambda_{C}$. In pattern recognition terminology, they are called *training data*. The samples used to build this model or template are called *biometric enrolment data record* (*enrolment data* in short from now on). The first sample captured is used for enrolment data, i.e., the first sample of the first session of each subject, as is usual in biometrics.**Cohort set** (impostor class training set), used to train the classifiers with examples of impostors. This set must be completely different from the impostor class test set in order to obtain objective results. Thus, we randomly split the individuals in the database different from the claimant into two different sets; one for training (cohort set), and the other for testing, as shown below. One sample is randomly selected from each individual in the cohort set. For objective comparisons, the cohort set so formed is always the same throughout all the experiments.

Once the subject model $\lambda_{C}$ had been trained, the tests were performed as shown in the following:
First, for each subject in the database, we selected the test samples:**Genuine test samples** (for biometric mated comparison trials [51]). For these tests, we used the two samples of the second session of the claimant.**Impostor test samples** (for biometric non-mated comparison trials). For these tests, we used *random forgeries*, i.e., a set of individuals in the database different from the claimant playing the role of impostors (system attackers); this is common in most biometric characteristics for technology evaluation, including gait, e.g., in [42,47], to cite two recent ones. For impostors, as mentioned, we used the subjects of the database different from the claimant not used in the *cohort set*. From each of these individuals, one of their samples was randomly selected to form this set, a set that is the same throughout all the experiments in order to achieve objective comparisons.For both genuine and impostor tests, the corresponding mated and non-mated trials for each subject *C* are accomplished from each test sample as follows:(a)The test sample is windowed, i.e., their cycles are grouped as shown in Section 3.3.(b)From each window, $w_{j}$, its corresponding feature vector, $F_{j}$, is extracted.(c)The corresponding score (classifier output), $s_{j}=s\left(F_{j}/\lambda_{C}\right)$, is calculated. This output is a *comparison score* [51].Therefore, for each test sample, we have as many comparison scores or test trials as windows into which it is divided.With these scores, two sets are created for each claimant *C*:One Test Set (TS) with genuine comparison scores, $TS_{g}^{C}$, achieved from the *genuine test samples*;Another test set with impostor comparison scores, $TS_{i}^{C}$, achieved from the *impostors test samples*.As reference, Table 2 shows the total number of tests performed for each window size, joining the corresponding test sets of all claimants.The system performance is calculated using these two sets, as shown in the next Section (Section 5.4).

When the WSFPP proposal is used, the sequence of steps is the same, except that a new one appears before the last. In this, the scores in $TS_{g}^{C}$ and $TS_{i}^{C}$ are fused in groups of *n*, as shown in Section 3.2. So, we have the following two sets:
One test set with genuine comparison scores, but now, these scores will be $s_{l}^{*}$. The set is achieved from the scores in $TS_{g}^{C}$. We call this set $TS_{g}^{*C}$.Another test set, $TS_{i}^{*C}$, with $s_{l}^{*}$ fused scores, but now achieved from $TS_{i}^{C}$.

### 5.4. Performance Measures

We evaluated Authentication performance using the False Match Rate (FMR, rate of impostor acceptance) and the False Non-Match Rate (FNMR, rate of claimant or genuine rejection). Since these measures are decision-threshold-dependent, graphical representations of the performance, such as the DET (Detection error trade-off) plot or the ROC (Receiver Operator Characteristic) curve, are generally used.

Nonetheless, using a single-number measure is more useful and easier to understand for a high number of comparisons. The Equal Error Rate (EER) is the most widely used in the biometrics literature. EER is the error of the system when the decision threshold is such that the FMR and FNMR are equal (in the ROC curve, the point where the diagonal cuts the curve). We used this measure here.

In order to obtain the final EER of the test, we calculated the EER for each subject *C* of the database, using $TS_{g}^{C}$ (genuine test scores) and $TS_{i}^{C}$ (impostor test scores) sets when the reference system was used, or using $TS_{g}^{*C}$ and $T_{i}^{*C}$ sets when our WSFPP approach was used. The final EER of the system is the mean of these EERs obtained from each subject.

## 6. Results

The results in the time domain can be seen in Figure 8, while those in the frequency domain are shown in Figure 9. The figures are organized into a matrix layout, where each row contains the results with the same classifier, and each column shows the results with the same sensor coordinate, including the *module*.

To better show and compare the data, a bar plot graph has been selected. Each bar plot contains the following information:
The title shows the feature extraction domain, the sensor coordinate (or fusing all by means of the module), and the classifier.The Y-axis shows the performance (EER in %). This axis has the same scale for each classifier to better compare results.The X-axis shows the results for each window size, measured by number of cycles.For each window size, four bars are shown. The first (brown) shows the result of the reference system (WSFPP is not used). The rest show the system performance when WSFPP is used, for n=2 (second bar, blue), n=4 (third bar, orange), and n=8 (fourth bar, purple).For each of these three last bars, the percentage of improvement or worsening with regard to the reference system (first bar) has been added; this calculation has been performed as shown in Equation (Equation 5), where $EER_{RefSys}$ is the performance of the reference system and $EER_{WSFPP}$ is the performance when WSFPP is used.
(5)$$(\left(EER_{RefSys}-EER_{WSFPP}\right)/EER_{RefSys})*100$$

The goal of using the proposed data visualization is to make the analysis easier. The performance comparisons of the classifier and sensor coordinates can be carried out without the necessity of “scrolling”, since the plots involved are on the same page: those comparisons related to the performance of the classifiers can be accomplished by comparing the results of one row with the others, while those related to the sensor coordinates can be accomplished by comparing the plots of a column with those in the other ones. Due to the great number of experiments, it has not been possible to put the results for time and frequency domains on the same page; however, in this case, these results are on consecutive pages, so the plots in the same position show the results with the same system configuration, except for the feature-extraction domain.

Focusing on each plot, the comparison of our proposal with regard to the reference system can be performed by comparing the first column of each group with the other three columns of the same group; each of these last three columns shows the results for a different value of the *n* (the number of fused window scores) parameter of our proposal. The analysis with regard to the window size is favored using the bar color; inside a plot, we can compare the results with the different window sizes tested, comparing the results of the bars with the same color, since only this parameter (window size) changes from one to another.

## 7. Discussion

From the results, the first important general conclusion is that our WSFPP proposal has improved the system performance in all of the scenarios, despite the differences in the system configurations tested. Therefore, the goal posed with the experimental study (Section 5.1) has been achieved, showing that the WSFPP technique is a successful proposal that can be widely used.

Details are provided as follows.

**With regard to the classifier.** Although the use of WSFPP has improved the results with all, this improvement is higher the better the performance of the classifier. The classifier with the best performance with the reference system is SVM, achieving improvements with WSFPP up to 90% in a lot of cases, even reaching 100%, which allowed 0% of EER to be achieved, a result not shown in any previous work. The second best classifier is Random Forest, which also achieves important improvements (higher than 90% in some cases) when WSFPP is used. The other two classifiers show a worse performance, and although the improvements are lower, these have reached 36% with 1-NN and 57% with MLP.**With regard to the feature extraction domain.** The state of the art shows mainly feature extraction in the time domain (Table 1). However, the results show that the features in the frequency domain are an interesting alternative, since similar, and sometimes even better, results have been achieved with these features. Focusing on the case when WSFPP is used, the frequency domain shows higher improvements in general, which has allowed 0% of EER to be reached; except for 1-NN, the best results were achieved in the frequency domain: 0.2% with Random Forest, 9% with MLP and 0% with SVM.**With regard to the window size.** There is no clear tendency. Both with the reference system and with WSFPP, the performance is dependent on the rest of the system parameters (sensor coordinates, feature extraction domain, and classifier). An interesting result is that, although not always, very good performances have been achieved with a size of two cycles, which is very small. Even more, with SVM and frequency domain features, 0% error has been achieved with this size, n=4 and X coordinates; this implies that, with a signal of only about 8 s, it has been possible to recognize a person by means of their way of walking using WSFPP.**With regard to the sensor coordinate.** The improvements with WSFPP are similar in all of the sensor coordinates, including the module: once the rest of the parameters of system have been fixed (each row in Figure 8 and Figure 9), the figures of the improvement are, in general, similar for the same values of *n*. This implies that the performance of WSFPP is independent of this parameter. Focusing on an analysis of the performance, as with the window size, it is dependent on the rest of the parameters of the system. However, if one must be selected, the best alternative is the module; the performance with this is, in the worst case, similar or slightly worst than the best with the other options (X, Y, or Z sensor coordinates), which are almost always better.**With regard to the value of *n* in the WSFPP proposal.** The first important aspect to note is that, with all of the values, the system performance has improved. This improvement is higher the higher the number of fused scores, *n*, is. Nevertheless, when the reference system has a good performance, very good results have been achieved with low values of *n*, e.g., with module, a window size of 2 cycles, frequency domain features and Random Forest (EER = 0.2% for n=4), or with the X coordinate, a window size of 2 cycles, frequency domain features, and SVM (EER = 0.05% for n=2).

As a summary, results from the conclusions that can be extracted from the above can be seen in Table 3: the best option is to use WSFPP with n=8, in the acquisition, to fuse the sensor coordinates by means of the module and to use features in the frequency domain except for 1-NN. In each case, the best window size is selected. Although the results in the table are not the best, they are representative of the improvements achieved by applying our WSFPP proposal to a state of the art system.

## 8. Conclusions

In this paper, a new proposal in biometric gait recognition, the Window Score Fusion post-processing technique, has been shown and successfully tested.

Following the state of the art, the proposal has been widely tested with different system configurations in all of the stages of the biometric system, with the goal of proving that the WSFPP proposal improves the recognition rates independently of the system configuration.

Improvements higher than 90% were achieved, e.g., 94% (from 3.2% to 0.2%, Figure 9e), with Random Forest or 100% (from 0.4% to 0%, Figure 9m) with SVM, have been achieved. In the worst cases (using K-NN and MLP as classifiers), the improvements achieved were not less than 30%, e.g., 36% (from 21.4% to 13.6%, Figure 8b) with K-NN, or 57% (from 20.1% to 8.6%, Figure 9l) with MLP.

From the results, it can be concluded that the proposed goal has been achieved, since our proposal has improved the recognition rates in all of the scenarios tested, showing that WSFPP is an interesting proposal that can be widely used.

The very good results achieved in user authentication, by means of biometric gait, allow us to predict a good performance of WSFPP in similar tasks. Among these, we propose, as interesting lines of future works, to approach identification with the same characteristic (gait) or recognition, in general, with other related biometric characteristics, for example, user recognition by means of Electrocardiogram (ECG), a characteristic that can also be captured by means of wearables.

## Figures and Tables

**Figure 1 sensors-23-01054-f001:**
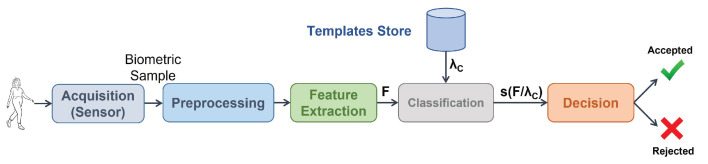
Main modules in a biometric system, where *F* stands for the feature vector (mathematical representation of the biometric sample), $\lambda_{C}$ stands for the template (model) of the user *C* to be authenticated (Claimant), and $s(F/\lambda_{C})$ is the score (classifier output), which is a measure of the degree of belonging of the biometric sample to the user.

**Figure 2 sensors-23-01054-f002:**
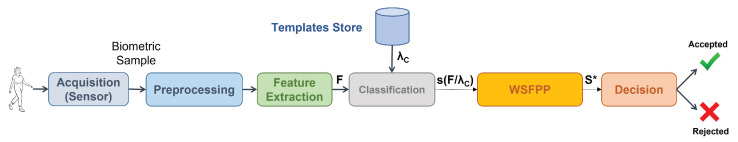
Proposal of system with Window Score Fusion Post-Processing (WSFPP) stage.

**Figure 3 sensors-23-01054-f003:**
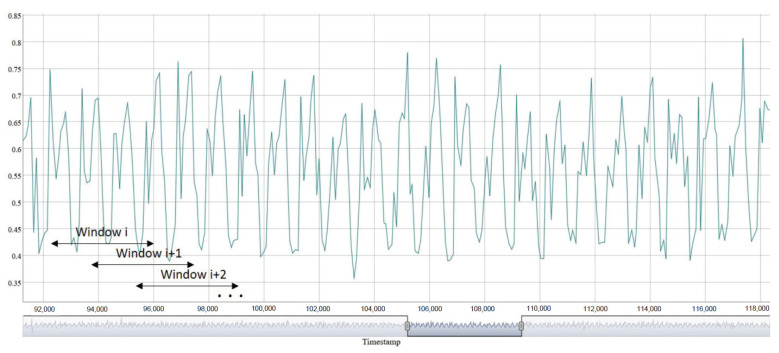
Example of sample windowing.

**Figure 4 sensors-23-01054-f004:**
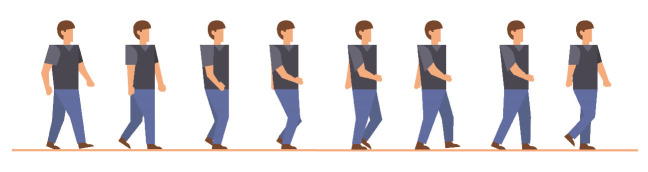
Gait cycle. Free image download from www.vecteezy.com (accessed on 4 November 2022).

**Figure 5 sensors-23-01054-f005:**
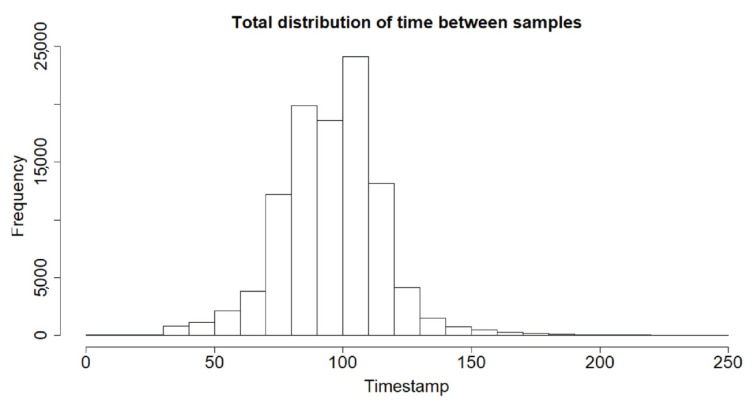
Histogram of the time interval between two consecutive datum. Timestamp is measured in milliseconds.

**Figure 6 sensors-23-01054-f006:**
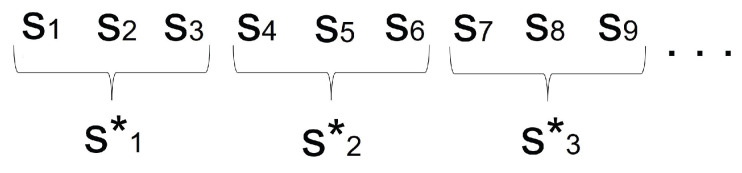
Example of window fusions at score level, for n=3. $s_{j}$ is the original score (classifier output for each sample window), while $s_{l}^{*}$ is the new score as a result of fusing *n* consecutive original scores.

**Figure 7 sensors-23-01054-f007:**
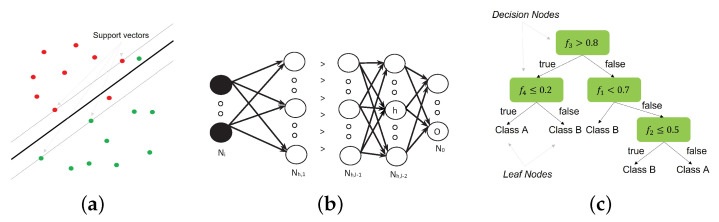
(**a**) SVM classifier. (**b**) MLP with l layers. (**c**) Decision tree example diagram.

**Figure 8 sensors-23-01054-f008:**
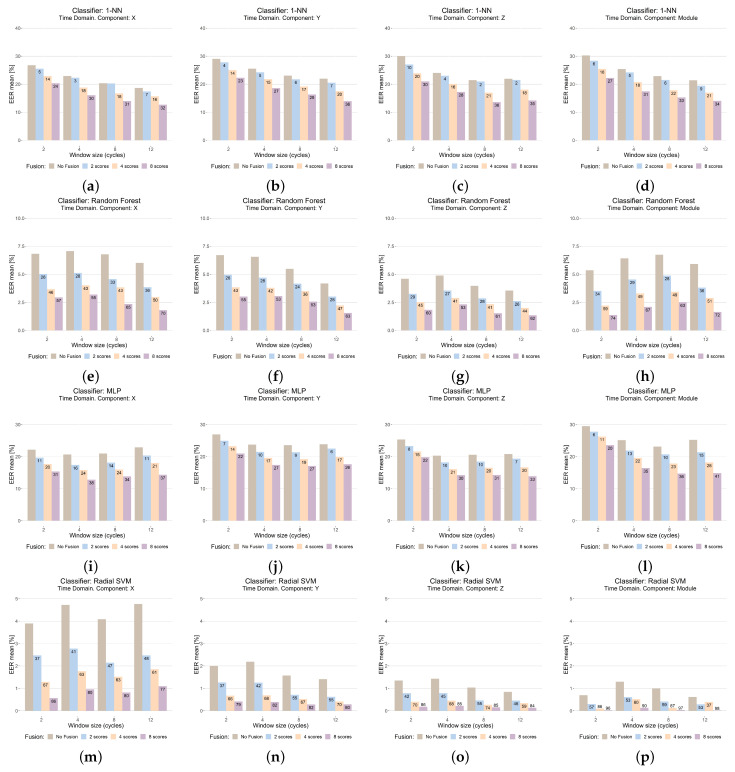
Results in the time domain.

**Figure 9 sensors-23-01054-f009:**
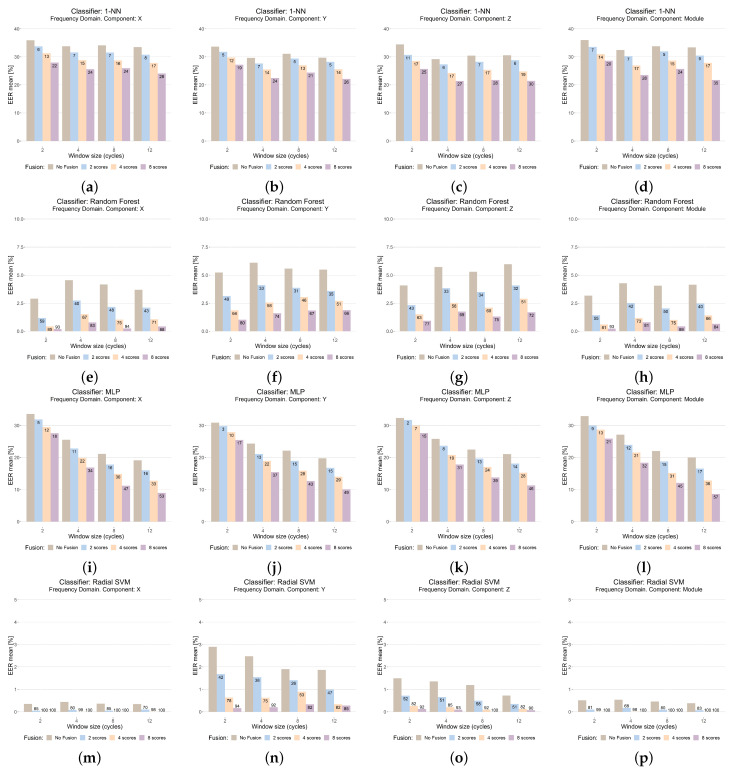
Results in the frequency domain.

**Table 1 sensors-23-01054-t001:** Brief description and best results with state-of-the-art proposals.

Work	Device	Classifier	Features	Window Size	Performance
Verma_22 [38]	WISDM-Database	Random Forest	Time domain: statistical, max-min value, time between peaks	10 s (10 cycles **, approximately)	EER *** = 11%
Vecchio_22 [40]	TicWatch E2	K-NN *	Based on [47]	Based on [47]	EER = 5%
Lee_22 [48]	Own-built wrist device	SVM	2D cyclogram features	Tested from 1 to 9 cycles	EER = 5.8%
Cola_21 [47]	Shimmer3	SVM *	Time domain: statistical and autocorrelation-based	Tested from 2 to 6 cycles (called gait segment)	EER = 3.5%
Giorgi_21 [41]	WISDM-Database	RNN *	raw data	2.56 s	EER = 2.4%
Kececi_20 [49]	Own-built	Ripper, MLP *, Random Forest, Decision Tree, k-NN, Bagging, Linear Regression, Random Tree, Naive Bayes, Bayesnet	Not found	Not found	FNMR = 0.3% FMR = 0.01%
Cheung_20 [42]	Smart-Watch	SVM	Time domain: statistical features	10-sample	EER = 6%
Weiss_19 [18]	Smart-watch	k-NN, Decision Tree, Random Forest	Time domain: statistical, max-min value, time between peaks	10 s (10 cycles, approximately)	EER = 6.8%
Musale_19 [44]	Smart-watch	Random Forest, K-NN, MLP	Time domain: statistical, correlation-based, physical (pitch, roll and yaw), force	Tested from 1 to 10 cycles	EER = 8.2%
Al-Naffakh_18 [19]	Smart-band	MLP	Time domain: statistical, correlation-based, max-min value, peaks-based	10 s (10 cycles, approximately)	EER = 0.05%
Wu_18 [50]	Own-built	SVM, ANN *, k-NN	Time domain (statistical, correlation, power, max-min) + Frequency domain (mean frequency, Bandwidth, Entropy) + Wavelet-domain (FFT Coefficient, Wavelet Energy)	Tested from 2 to 11 s (cycles, approximately)	FNMR = 5.0% FMR = 4.7%
Xu_17 [45]	Smart-watch	Sparse Fusion	Sparse Fusion Classification	Tested from 1 to 6 cycles for identification task and fixed to 8 cycles for verification task	EER = 3.1%
Johnston_15 [21]	Smart-watch	MLP, Random Forest, Rotation Forest, Naive Bayes	Time domain: statistical, time between peaks, max-min	10 s (10 cycles, approximately)	EER = 1.4%

* SVM: Support Vector Machine. MLP: Multilayer Perceptron. k-NN: K Nearest Neighbor. ANN: Autoencoder Neural Network. RNN: Recurrent Neural Network. ** Cycle is defined in Section 3 and Window (cycles set) in Section 3.3. *** Performance measures are defined in Section 5.4.

**Table 2 sensors-23-01054-t002:** Test set sizes for the different window sizes tested. Columns #Genuine and #Impostor show the number of genuine and impostor test trials, respectively.

Window Size	#Genuine	#Impostor
2 cycles	11,353	224,650
4 cycles	5486	108,285
8 cycles	2632	51,701
12 cycles	1729	33,824

**Table 3 sensors-23-01054-t003:** Results summary. Column *WS* shows the window size, *RF* the result with the Reference System, and *WSFPP* when our proposal is used with n=12. *Module* is used in the acquisition stage and frequency domain in the feature extraction stage, except for 1-NN, which performs better in the time domain.

Classifier	WS	RF	WSFPP
1-NN	12 cycles	21.4%	14%
Random Forest	2 cycles	3.2%	0.2%
MLP	12 cycles	20.1%	8.6%
SVM	12 cycles	0.4%	0%

## Data Availability

Not applicable.

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
