# Peer review of "A New Post-Processing Proposal for Improving Biometric Gait Recognition Using Wearable Devices"

_sensors, 2023, doi:10.3390/s23031054_

Round 1
Reviewer 1 Report
The authors have done a good piece of research job on behavioral biometric system conducting wearable devices to improve the gait recognition.
The experimental results have proved the authors' approach. However, the methodology is only seen from an experimental point of view. The theory was dealt with only when showing the experimental design, but still without a clear mathematical model or at least an algorithmic description.
A clear theoretical model would enhance the work and give the sense of novelty, particularly when compared with the state of the art, which in turn is given in a good detail by the authors.
Moreover, I would suggest giving more comparison results in the conclusion section.
Reviewer 2 Report
The authors propose a post-processing technique for gait recognition and test it with different configurations. The main recommendations are as follows:
11) As the representative of processing time series data in the current popular deep learning model, long short term memory (LSTM) can better process time series data including gait, which has been proved in a large number of existing literature. It is recommended to add such classifiers when comparing experimental results.
22) The references are outdated, and the research on the latest gait research literature is insufficient, which is difficult to prove the significance and innovation of this method. It is suggested to add the research literature of the last three years.
Reviewer 3 Report
The manuscript offers unique insight on gait analysis with the use of wearables that can be of great interest to the readership. There are areas where the manuscript has potential to better serve the reader.
[Lines 1-10] Abstract is critical to the paper and the idea that you are introducing a post-processing technique is clear, but in both the title and the abstract you introduce “Window Score Fusion” that confounds the clarity of simply stating “new post-processing technique.” This phrase in the title puts off the reader a bit since the title is so very long.
[Lines 505-509] You indicate the algorithm is “tested” in the abstract and not clinically validated. To improve standard of care for biomedical applications it would be helpful to clarify. You indicate that the study did not require IRB approval, but them hint in line 506 that you offered consent. Was this study ruled exempt by the IRB or was that self-determined. This plays significantly into the credibility of the paper.
[Lines 33-35] Please clarify “dedicated devices” -- the smartwatch is a wearable and does not require the “surroundable” camera systems. Inconvenient, but the gold standard. Do the surroundable system require ML? Are you trading one compromise for another?
[Lines 54-55] Is the fact that a device is “popular” a solid technical foundation for this study?
[Line 60-63] It is not self-evident what you mean by “system proposed” for the standard biometric method; is it the traditional system proposed or the system you are proposing included? Is there a fundamental reason that poor pre-processing will be corrected by enhanced post-processing? Please expand Figure 5; I think this is an important aspect of windowing, but the message seems lost about fusion and windowing.
[Lines 78-79] Is the sentence complete? I had to read a few times and the message is lost on me.
[Line 86] Please clarify. It is hard to tell if these are pre-existing proposals or new proposals related to your research.
[Lines105-107] It seems as if you have identified a gap in the technology of signal processing, but does the gap itself warrant research absent a more compelling reason? In other words, have you identified specific issues that can be better fixe with pre- versus post-processing?
[Lines 121-132] Excellent overview of the data in Table 1.
[Lines 160-186] Here is where you might identify specific deficiencies. On line 177 you indicate what is “reasonable” – is there a metric for this?
[Lines 224-226] So is the required training an asset to the analytics in your case? Of general interest would be how you fine-tune pre-processing if you are claiming that post-processing improves outcomes.
[Line 282] For kNN you have picked k=1? Were the previous experiments those that you conducted?
[Lines 247++] Methodology section is clear, thank you!
[Lines 404-423] Results. The plots are very difficult to read and it is hard to understand the points they are trying to make. The results do not convey significance of the outcome. Some additional narrative is required to underscore what is important. You say the “figure shows” but it is not compelling.
[Line 478-484] It is very difficult to endorse the success without summative detail (see prior notes) and care with tested versus validated remains confusing
Round 2
Reviewer 1 Report
Thank you for the sufficient incorporation of my comments.
Reviewer 2 Report
The authors have addressed all my concerns. I don't have any more questions.
Reviewer 3 Report
Thank you for your efforts on the revised paper. Many facets are much clearer and easier to follow.